# Online Learning for Prediction via Covariance Fitting: Computation, Performance and Robustness

**Muhammad Osama**                                  *muhammad.osama@it.uu.se*
*Department of Information Technology,*
*Uppsala University*

**Dave Zachariah**                                  *dave.zachariah@it.uu.se*
*Department of Information Technology,*
*Uppsala University*

**Petre Stoica**                                    *ps@it.uu.se*
*Department of Information Technology,*
*Uppsala University*

**Thomas B. Schön**                                 *thomas.schon@it.uu.se*
*Department of Information Technology,*
*Uppsala University*

**Reviewed on OpenReview:** *https://openreview.net/forum?id=nAr9PhyEbQ*

## Abstract

We consider the problem of online prediction using linear smoothers that are functions of a nominal covariance model with unknown parameters. The model parameters are often learned using cross-validation or maximum-likelihood techniques. But when training data arrives in a streaming fashion, the implementation of such techniques can only be done in an approximate manner. Even if this limitation could be overcome, there appears to be no clear-cut results on the statistical properties of the resulting predictor.

Here we consider a covariance-fitting method to learn the model parameters, which was initially developed for spectral estimation. We first show that the use of this approach results in a computationally efficient online learning method in which the resulting predictor can be updated sequentially. We then prove that, with high probability, its out-of-sample error approaches the optimal level at a root-$n$ rate, where $n$ is the number of data samples. This is so even if the nominal covariance model is misspecified. Moreover, we show that the resulting predictor enjoys two robustness properties. First, it corresponds to a predictor that minimizes the out-of-sample error with respect to the least favourable distribution within a given Wasserstein distance from the empirical distribution. Second, it is robust against errors in the covariate training data. We illustrate the performance of the proposed method in a numerical experiment.

## 1 Introduction

We consider scenarios in which we observe a *stream* of randomly distributed data

$$\mathcal{D}_n = \{(\mathbf{x}_1, y_1), \ldots, (\mathbf{x}_n, y_n)\} \qquad n = 1, 2, 3, \ldots$$

Given covariate $\mathbf{x}_{n+1}$ in a space $\mathcal{X}$, our goal is to predict the outcome $y_{n+1}$ in $\mathbb{R}$. A large class of predictors (also known as linear smoothers) can be described as a weighted combination of observed outcomes:

$$\widehat{y}(\mathbf{x}; \boldsymbol{\lambda}) = \sum_{i=1}^{n} w_i(\mathbf{x}; \boldsymbol{\lambda}) y_i, \tag{1}$$

where $\mathbf{x}$ denotes any test point and the weights $\{w_i(\mathbf{x}; \boldsymbol{\lambda})\}$ are parameterized by $\boldsymbol{\lambda}$. The sensitivity of such a predictor function to noise in the training data is often characterized by how close the in-sample prediction $\widehat{y}(\mathbf{x}_i; \boldsymbol{\lambda})$ is to $y_i$ and quantified by the sum of in-sample weights,

$$0 < \; df_n \triangleq \sum_{i=1}^{n} w_i(\mathbf{x}_i; \boldsymbol{\lambda}), \tag{2}$$

also known as the 'effective' degrees of freedom (Ruppert et al., 2003; Wasserman, 2006; Hastie et al., 2009). These degrees of freedom are often tuned to avoid overfitting to the irreducible noise in the training data with the aim of achieving good out-of-sample performance. This includes learning the parameters $\boldsymbol{\lambda}$ from $\mathcal{D}_n$ via distribution-free cross-validation or distribution-based maximum likelihood methods, which however can typically be implemented only approximately in the online scenario.

In this paper, we consider an alternative method using a covariance-based criterion first proposed in the context of spectral estimation (Stoica et al., 2010a;b). We show that this method

- enables sequential computation of a predictor with learned parameters,
- approaches an optimal out-of-sample performance at a root-$n$ rate,
- enjoys two types of robustness properties.

For illustration of the online learning method, we include a numerical experiment.

*Notation:* $\|\mathbf{Z}\|_{\mathbf{W}} = \sqrt{\mathrm{tr}\{\mathbf{Z}^\top \mathbf{W} \mathbf{Z}\}}$ is a weighted Frobenious norm of matrix $\mathbf{Z}$ using a positive definite weight matrix $\mathbf{W}$. The element-wise Hadamard product between $\mathbf{z}$ and $\mathbf{z}'$ is denoted $\mathbf{z} \odot \mathbf{z}'$.

## 2 Problem formulation

The linear smoother predictor (1) can be written compactly as

$$\widehat{y}(\mathbf{x}; \boldsymbol{\lambda}) = \mathbf{w}^\top(\mathbf{x}; \boldsymbol{\lambda})\mathbf{y}, \quad \text{where } \mathbf{w}^\top(\mathbf{x}; \boldsymbol{\lambda}) = [w(\mathbf{x}_1; \boldsymbol{\lambda}), \dots, w(\mathbf{x}_n; \boldsymbol{\lambda})] \text{ and } \mathbf{y} = [y_1, \dots, y_n]^\top. \tag{3}$$

We will investigate a class of model-based weights for which (3) can be computed sequentially from the stream $\mathcal{D}_n$. Specifically, suppose $y$ is modeled as a zero-mean stochastic process with a nominal covariance function parameterized as

$$\mathrm{Cov}[y, y' | \mathbf{x}, \mathbf{x}'; \boldsymbol{\lambda}] = \lambda_0 \delta(\mathbf{x}, \mathbf{x}') + \sum_{k=1}^{d} \lambda_k \phi_k(\mathbf{x}) \phi_k(\mathbf{x}'), \tag{4}$$

where $\mathbf{x}$ and $\mathbf{x}'$ are two arbitrary covariates, $\delta(\mathbf{x}, \mathbf{x}')$ is the Kronecker delta function and $\{\phi_k(\mathbf{x})\}$ are real-valued features of $\mathbf{x}$. Here $\boldsymbol{\lambda}$ is the *unknown* set of $d + 1$ nonnegative model covariance parameters. When $\mathbf{x}$ belongs to a vector space, then periodic Fourier-type features provide a convenient choice due to their excellent covariance approximating properties (Ruppert et al., 2003; Rahimi & Recht, 2007; Stein, 2012; Hensman et al., 2017; Solin & Särkkä, 2020). Using the notation above, a set of optimal weights can be written as

$$\mathbf{w}(\mathbf{x}; \boldsymbol{\lambda}) = \mathbf{C}_{\boldsymbol{\lambda}}^{-1} \boldsymbol{\Phi} \boldsymbol{\Lambda} \boldsymbol{\phi}(\mathbf{x}), \tag{5}$$

(Bishop, 2006; Rasmussen & Williams, 2006; Stein, 2012) where

$$\boldsymbol{\phi}(\mathbf{x}) = \begin{bmatrix} \phi_1(\mathbf{x}) \\ \vdots \\ \phi_d(\mathbf{x}) \end{bmatrix}, \qquad \boldsymbol{\Phi} = \begin{bmatrix} \boldsymbol{\phi}^\top(\mathbf{x}_1) \\ \vdots \\ \boldsymbol{\phi}^\top(\mathbf{x}_n) \end{bmatrix}$$

contain the features and

$$\mathbf{C}_{\boldsymbol{\lambda}} = \boldsymbol{\Phi}\boldsymbol{\Lambda}\boldsymbol{\Phi}^{\top} + \lambda_0 \mathbf{I}_n \succ \mathbf{0} \tag{6}$$

is a nominal covariance matrix where $\boldsymbol{\Lambda} = \mathrm{diag}(\lambda_1, \ldots, \lambda_d)$. The predictor function above includes a variety of penalized regression methods (see the references cited above). The degrees of freedom of (3) are controlled by $\boldsymbol{\lambda}$ and we have that

$$0 < df_n(\boldsymbol{\lambda}) = \mathrm{tr}\left\{\boldsymbol{\Phi}\boldsymbol{\Lambda}\boldsymbol{\Phi}^{\top}\mathbf{C}_{\boldsymbol{\lambda}}^{-1}\right\} = \mathrm{tr}\left\{\boldsymbol{\Phi}\boldsymbol{\Lambda}\boldsymbol{\Phi}^{\top}(\boldsymbol{\Phi}\boldsymbol{\Lambda}\boldsymbol{\Phi}^{\top} + \lambda_0 \mathbf{I}_n)^{-1}\right\} \leq \min(n, d) \tag{7}$$

(Stoica & Stanasila, 1982; Ruppert et al., 2003).

Let $\boldsymbol{\lambda}_n$ denote the parameters that are *fitted* to the data stream $\mathcal{D}_n$ in some way. The concern of this paper is two-fold: (i) *sequential* computation of $\widehat{y}(\mathbf{x}; \boldsymbol{\lambda}_n)$ and (ii) performance *guarantees* that hold even when the model class is not well-specified. That is, the nominal covariance model (4) using $\{\boldsymbol{\phi}_k(\mathbf{x})\}$ may not match the unknown covariance structure of the data. Nevertheless, we aim to derive meaningful guarantees that remain valid also in this misspecified case.

Cross-validation or maximum likelihood methods are two popular approaches to fitting $\boldsymbol{\lambda}_n$. However, these methods have challenges in the online learning setting. First, computing $\boldsymbol{\lambda}_n$ for the covariance model above is a non-convex problem that can be riddled with multiple local minima. Second, for each additional training data point $(\mathbf{x}_{n+1}, y_{n+1})$, the parameters $\boldsymbol{\lambda}_{n+1}$ will have to be refitted and therefore $\widehat{y}(\mathbf{x}; \boldsymbol{\lambda}_n)$ recomputed from scratch or approximated (see below). Third, to the best of our knowledge, finite out-of-sample prediction performance guarantees for $\widehat{y}(\mathbf{x}; \boldsymbol{\lambda}_n)$ neither exist for cross-validation nor maximum likelihood $\boldsymbol{\lambda}_n$.

We will consider an alternative covariance-based fitting criterion for $\boldsymbol{\lambda}$, used in another context, viz. spectral estimation (Stoica et al., 2010a;b). The main contribution is to derive performance guarantees of a sequentially computable predictor, thereby establishing predictive properties of the covariance-fitting methodology. Similar results do not appear to be available in the previous literature.

## 3 Other approaches to online model-based prediction

We identify two main lines of work that enable online or fast implementations of model-based linear smoothers, which include Gaussian process regression and kriging methods: The first line considers spectral model approximations (e.g., (Rahimi & Recht, 2007; Hensman et al., 2017; Solin & Särkkä, 2020)) which is covered by the class of covariance models in (4). These methods also enable efficient online computation of $\widehat{y}(\mathbf{x}; \boldsymbol{\lambda})$, but for a fixed set of model parameters $\boldsymbol{\lambda}$. The second line considers sparse variational approximations (e.g., (Titsias, 2009; Bui et al., 2017; Stanton et al., 2021)). These methods can recompute the predictor $\widehat{y}(\mathbf{x}; \boldsymbol{\lambda})$ efficiently when new data arrives, but again for fixed $\boldsymbol{\lambda}$.

Fitting $\boldsymbol{\lambda}_n$ and computing $\widehat{y}(\mathbf{x}; \boldsymbol{\lambda}_n)$ in a sequential manner requires recomputing past covariance quantities and is computationally prohibitive. For the covariance model above, it means recomputing (5) and the inverse of (6) at *each new sample*. Work on this problem appears to be scarce. Stanton et al. (2021) consider seeking maximum likelihood estimates $\boldsymbol{\lambda}_n$ by a type of gradient-based search that projects past quantities onto a lower-dimensional space. However, neither a convergence analysis towards the sought maximum likelihood estimate nor any resulting predictive properties are provided. By contrast, Cai & Yuan (2012) provide an asymptotic performance guarantee of a linear smoother with a learned regularization parameter, but their approach is restricted to the offline setting.

Our focus below is on deriving finite-sample performance guarantees for a sequentially computable predictor.

## 4 Learning via covariance fitting

The parameters $\boldsymbol{\lambda}$ can be learned by fitting the model covariance matrix $\mathbf{C}_{\boldsymbol{\lambda}}$ in (5) to the empirical covariance matrix $\mathbf{y}\mathbf{y}^{\top}$. Specifically, we will use the following fitting criterion, known as SPICE (Stoica et al., 2010a;b),

$$\boxed{\boldsymbol{\lambda}_n = \underset{\boldsymbol{\lambda} \geq \mathbf{0}}{\arg\min} \; \left\|\mathbf{y}\mathbf{y}^{\top} - \mathbf{C}_{\boldsymbol{\lambda}}\right\|_{\mathbf{C}_{\boldsymbol{\lambda}}^{-1}}^2,} \tag{8}$$

where $\boldsymbol{\lambda}_n$ is a function of the datastream $\mathcal{D}_n$. Since this criterion is convex in $\boldsymbol{\lambda}$, we can be sure that a global minimizer can be determined.

## 4.1 Sequential computation

The covariance-fitting criterion in (8) can be connected to an alternative convex problem and this connection can be utilized to perform parameter estimation online, as shown in (Zachariah & Stoica, 2015). We first leverage this connection to compute $\widehat{y}(\mathbf{x}; \boldsymbol{\lambda}_n)$ in a sequential manner, and subsequently we derive a series of predictive performance guarantees of the proposed method.

**Theorem 1** *The predictor function $\widehat{y}(\mathbf{x}; \boldsymbol{\lambda}_{n+1})$ can be updated from $\widehat{y}(\mathbf{x}; \boldsymbol{\lambda}_n)$ in a constant runtime $\mathcal{O}\left(d^2\right)$. The total memory requirement of the method is also on the order of $\mathcal{O}\left(d^2\right)$.*

**Proof 1** *We first note that the predictor (3) has an equivalent form*

$$\widehat{y}(\mathbf{x}; \boldsymbol{\lambda}) = \boldsymbol{\phi}^\top(\mathbf{x}) \underbrace{\boldsymbol{\Lambda}\boldsymbol{\Phi}^\top\mathbf{C}_{\boldsymbol{\lambda}}^{-1}\mathbf{y}}_{\widehat{\boldsymbol{\theta}}(\boldsymbol{\lambda})} \tag{9}$$

*For later use, we also note that (9) is invariant to a uniform rescaling of $\boldsymbol{\lambda}$, since*

$$\widehat{\boldsymbol{\theta}}(\boldsymbol{\lambda}) \equiv \widehat{\boldsymbol{\theta}}(\alpha\boldsymbol{\lambda}), \quad \forall \alpha > 0 \tag{10}$$

*The covariance-fitting criterion in (8) can be expanded into*

$$\left\|\mathbf{y}\mathbf{y}^\top - \mathbf{C}_{\boldsymbol{\lambda}}\right\|_{\mathbf{C}_{\boldsymbol{\lambda}}^{-1}}^2 = \mathbf{y}^\top\mathbf{C}_{\boldsymbol{\lambda}}^{-1}\mathbf{y} \cdot \|\mathbf{y}\|^2 + tr\{\mathbf{C}_{\boldsymbol{\lambda}}\} + K, \tag{11}$$

*where $K$ is a constant. Thus $\boldsymbol{\lambda}_n$ is also a minimizer of*

$$V(\boldsymbol{\lambda}) = \mathbf{y}^\top\mathbf{C}_{\boldsymbol{\lambda}}^{-1}\mathbf{y} + \|\mathbf{y}\|^{-2} \cdot tr\{\mathbf{C}_{\boldsymbol{\lambda}}\}$$

*Next, we follow Zachariah & Stoica (2015, Appendix A) and consider the following augmented convex criterion,*

$$V'(\boldsymbol{\theta}, \boldsymbol{\lambda}) = \frac{1}{\lambda_0}\|\mathbf{y} - \boldsymbol{\Phi}\boldsymbol{\theta}\|_2^2 + \boldsymbol{\theta}^\top\boldsymbol{\Lambda}^{-1}\boldsymbol{\theta} + tr\{\mathbf{C}_{\boldsymbol{\lambda}}\}, \tag{12}$$

*and show that its minimizers produce the predictor $\widehat{y}(\mathbf{x}; \boldsymbol{\lambda}_n) = \boldsymbol{\phi}^\top(\mathbf{x})\widehat{\boldsymbol{\theta}}(\boldsymbol{\lambda}_n)$.*

*The first argument of (12) is minimized by*

$$\widehat{\boldsymbol{\theta}}(\boldsymbol{\lambda}) = \left(\lambda_0^{-1}\boldsymbol{\Phi}^\top\boldsymbol{\Phi} + \boldsymbol{\Lambda}^{-1}\right)^{-1}\boldsymbol{\Phi}^\top\lambda_0^{-1}\mathbf{y} = \boldsymbol{\Lambda}\boldsymbol{\Phi}^\top\mathbf{C}_{\boldsymbol{\lambda}}^{-1}\mathbf{y},$$

*where the second equality follows from using the matrix inversion lemma. Inserting the minimizer it back into (12), we have a concentrated cost function:*

$$V'(\widehat{\boldsymbol{\theta}}(\boldsymbol{\lambda}), \boldsymbol{\lambda}) = \mathbf{y}^\top\mathbf{C}_{\boldsymbol{\lambda}}^{-1}\mathbf{y} + tr\{\mathbf{C}_{\boldsymbol{\lambda}}\}$$

*Let us now consider the minimizing $\boldsymbol{\lambda}$. By rescaling the parameters by $\alpha = \|\mathbf{y}\|^{-1} > 0$, we have that*

$$\begin{aligned}\alpha \cdot V'(\widehat{\boldsymbol{\theta}}(\alpha\boldsymbol{\lambda}), \alpha\boldsymbol{\lambda}) &= \alpha \cdot \left(\mathbf{y}^\top(\alpha\mathbf{C}_{\boldsymbol{\lambda}})^{-1}\mathbf{y} + \cdot tr\{\alpha\mathbf{C}_{\boldsymbol{\lambda}}\}\right) \\ &= V(\boldsymbol{\lambda})\end{aligned}$$

*where $\alpha = \|\mathbf{y}\|^{-1} > 0$. It follows that $\alpha\boldsymbol{\lambda}_n$ is a minimizer of (12), since*

$$V'(\widehat{\boldsymbol{\theta}}(\alpha\boldsymbol{\lambda}_n), \alpha\boldsymbol{\lambda}_n) = \frac{1}{\alpha}V(\boldsymbol{\lambda}_n) \leq \frac{1}{\alpha}V(\boldsymbol{\lambda}) = V'(\widehat{\boldsymbol{\theta}}(\alpha\boldsymbol{\lambda}), \alpha\boldsymbol{\lambda}) \qquad \forall\boldsymbol{\lambda} \geq \mathbf{0}.$$

This scaled minimizer can be related to the predictor with fitted parameters: $\widehat{y}(\mathbf{x}; \boldsymbol{\lambda}_n) = \boldsymbol{\phi}^\top(\mathbf{x})\widehat{\boldsymbol{\theta}}(\boldsymbol{\lambda}_n) \equiv \boldsymbol{\phi}^\top(\mathbf{x})\widehat{\boldsymbol{\theta}}(\alpha\boldsymbol{\lambda}_n)$ using (10).

We now change the order of the minimization of $V'(\boldsymbol{\theta}, \boldsymbol{\lambda})$ to arrive at an alternative way of computation. The minimizing parameters are

$$\lambda_k(\boldsymbol{\theta}) = \begin{cases} \frac{\|y - \boldsymbol{\Phi}\boldsymbol{\theta}\|_2}{\sqrt{n}}, & k = 0 \\ \frac{|\theta_i|}{\sqrt{n}\psi_k}, & k = 1, \ldots, d \end{cases} \tag{13}$$

where $\psi_k = \sqrt{\frac{1}{n}\sum_{i=1}^n \phi_k^2(\mathbf{x}_i)}$. Inserting (13) into (12) yields the following equivalent convex cost function

$$\arg\min_{\boldsymbol{\theta}} \sqrt{\frac{1}{n}\|\mathbf{y} - \boldsymbol{\Phi}\boldsymbol{\theta}\|_2^2} + \frac{1}{\sqrt{n}}\|\boldsymbol{\psi} \odot \boldsymbol{\theta}\|_1, \tag{14}$$

where $\boldsymbol{\psi} = [\psi_1 \cdots \psi_d]^\top$. Let $\boldsymbol{\theta}_n$ denote the minimizer of (14), then $\widehat{y}(\mathbf{x}; \boldsymbol{\lambda}_n) \equiv \boldsymbol{\phi}^\top(\mathbf{x})\boldsymbol{\theta}_n$.

Eq. (14) is a weighted square-root LASSO problem (Belloni et al., 2011) that can be solved in a runtime on the order of $\mathcal{O}(d^2)$ using variables

$$\mathbf{A}^n = \sum_{i=1}^n \boldsymbol{\phi}(\mathbf{x}_i)\boldsymbol{\phi}^\top(\mathbf{x}_i), \quad \mathbf{b}^n = \sum_{i=1}^n \boldsymbol{\phi}^\top(\mathbf{x}_i)y_i, \quad c^n = \sum_{i=1}^n y_i^2, \tag{15}$$

of fixed dimension that are updated recursively. Thus the memory requirement is dominated by the storing of the $d \times d$-matrix $\mathbf{A}^n$.

The pseudocode for the method is provided in the appendix.

## 4.2   Out-of-sample performance

We now turn to evaluating the out-of-sample performance of the predictor learned from the data stream $\mathcal{D}_n$. Specifically, we consider the mean-squared error

$$\textsc{Mse} = \mathbb{E}\left[\left(y_{n+1} - \widehat{y}(\mathbf{x}_{n+1}; \boldsymbol{\lambda}_n)\right)^2\right], \tag{16}$$

for the *subsequent* sample $(\mathbf{x}_{n+1}, y_{n+1})$ in the stream. The expectation is conditional on $\mathcal{D}_n$, thus the Mse will depend on the particular realization of the stream since the learned predictor is a function of all past samples.

To provide a performance reference, we note that all predictors of the form (3) with (5) belong to the following class of predictor functions

$$\mathcal{F} \triangleq \left\{ f(\mathbf{x}) = \sum_{k=1}^d \phi_k(\mathbf{x})\theta_k \; : \; \boldsymbol{\theta} \in \mathbb{R}^d \right\}, \tag{17}$$

i.e., a linear combination of all features $\{\phi_k(\mathbf{x})\}$ in the nominal model (4). We can now benchmark $\widehat{y}(\mathbf{x}; \boldsymbol{\lambda}_n)$ against the minimal achievable error among all predictors in $\mathcal{F}$, even when the nominal covariance model (4) is misspecified. Specifically, we provide the following finite out-of-sample performance guarantee for the learned predictor.

**Theorem 2** *Assume the outcome $y$ and features are $\phi(\mathbf{x})$ bounded, and that the features are such that $\sum_{i=1}^n |\phi_k(\mathbf{x}_i)|^2 > 0$. If the data pairs $(\mathbf{x}_i, y_i)$ in the stream are drawn i.i.d., then the out-of-sample error of $\widehat{y}(\mathbf{x}; \boldsymbol{\lambda}_n)$ is given by*

$$\boxed{\mathbb{E}\left[\left(y_{n+1} - \widehat{y}(\mathbf{x}_{n+1}; \boldsymbol{\lambda}_n)\right)^2\right] \leq \min_{\widehat{y}\in\mathcal{F}} \mathbb{E}\left[\left(y_{n+1} - \widehat{y}(\mathbf{x}_{n+1})\right)^2\right] + K\sqrt{\frac{1}{n}\ln\frac{2(d+1)^2}{\varepsilon}} + b_n} \tag{18}$$

*with probability of at least $1 - \varepsilon$, where $K$ is a constant and $b_n$ is bounded as $\mathcal{O}(n^{-3/4})$. That is, with high probability, the out-of-sample error approaches the minimum achievable error at a root-n rate. Note that the number of features $d$ increases only the second term at a logarithmic rate.*

**Proof 2** *For notational simplicity, let $(\mathbf{x}, y)$ denote the random $(n+1)$th sample. Let $\widehat{y}$ be any predictor in $\mathcal{F}$ and express its out-of-sample mean-square error as:*

$$R(\widehat{y}) \equiv \mathbb{E}\left[\left(y - \boldsymbol{\phi}^\top(\mathbf{x})\boldsymbol{\theta}\right)^2\right] = \mathbb{E}\left[\begin{bmatrix} \boldsymbol{\theta} \\ -1 \end{bmatrix}^\top \underbrace{\begin{bmatrix} \boldsymbol{\phi}(\mathbf{x}) \\ y \end{bmatrix}}_{\mathbf{z}} \begin{bmatrix} \boldsymbol{\phi}(\mathbf{x}) \\ y \end{bmatrix}^\top \underbrace{\begin{bmatrix} \boldsymbol{\theta} \\ -1 \end{bmatrix}}_{\widetilde{\boldsymbol{\theta}}}\right] = \widetilde{\boldsymbol{\theta}}^\top \boldsymbol{\Sigma} \widetilde{\boldsymbol{\theta}}, \tag{19}$$

*where $\boldsymbol{\Sigma} = \mathbb{E}[\mathbf{z}\mathbf{z}^\top]$. Similarly, the in-sample error can be expressed as $R_n(\widehat{y}) = \widetilde{\boldsymbol{\theta}}^\top \widehat{\boldsymbol{\Sigma}} \widetilde{\boldsymbol{\theta}}$, where $\widehat{\boldsymbol{\Sigma}} = n^{-1}(\mathbf{z}_1\mathbf{z}_1^\top + \cdots + \mathbf{z}_n\mathbf{z}_n^\top)$. The gap between in- and out-of-sample errors can be bounded as:*

$$\begin{aligned} |R_n(\widehat{y}) - R(\widehat{y})| &= |\widetilde{\boldsymbol{\theta}}^\top (\widehat{\boldsymbol{\Sigma}} - \boldsymbol{\Sigma})\widetilde{\boldsymbol{\theta}}| \\ &\leq \sum_{i=1}^{d+1}\sum_{j=1}^{d+1} |\widetilde{\theta}_i\|\widetilde{\theta}_j|\,|\widehat{\boldsymbol{\Sigma}}_{ij} - \boldsymbol{\Sigma}_{ij}| \\ &\leq (\|\boldsymbol{\theta}\|_1 + 1)^2 \cdot \underbrace{\max_{i,j}\,|\widehat{\boldsymbol{\Sigma}}_{ij} - \boldsymbol{\Sigma}_{ij}|}_{\widetilde{\sigma}} \end{aligned} \tag{20}$$

*Next, we bound $\widetilde{\sigma}$ (see also Greenshtein & Ritov (2004)). Since $y$ and $\boldsymbol{\phi}(\mathbf{x})$ are bounded random variables, we have that $|z_i z_j| \leq B$ for some $B$ and using Hoeffding's inequality*

$$\Pr\left\{|\widehat{\boldsymbol{\Sigma}}_{ij} - \boldsymbol{\Sigma}_{ij}| \geq \sigma\right\} \leq 2\exp\left(-\frac{n\sigma^2}{2B^2}\right) \tag{21}$$

*Combining this result with the union bound over all $(d+1)^2$ variables in $\widetilde{\sigma}$, we have that*

$$\Pr\left\{\widetilde{\sigma} \geq \sigma\right\} \leq (d+1)^2 \cdot 2\exp\left(-\frac{n\sigma^2}{2B^2}\right) \triangleq \varepsilon \tag{22}$$

*Consequently, we can replace $\widetilde{\sigma}$ by*

$$\sigma = B\sqrt{\frac{2}{n}}\sqrt{\ln\frac{2(d+1)^2}{\varepsilon}} \tag{23}$$

*in (20) so that*

$$|R_n(\widehat{y}) - R(\widehat{y})| \leq (\|\boldsymbol{\theta}\|_1 + 1)^2\sigma,$$

*holds for any predictor in $\mathcal{F}$ with a probability of at least $1 - \varepsilon$. Thus if $\boldsymbol{\theta}$ is bounded, $\|\boldsymbol{\theta}\|_1 \leq P$ for some $P$, then*

$$R(\widehat{y}) - (P+1)^2\sigma \leq R_n(\widehat{y}) \leq R(\widehat{y}) + (P+1)^2\sigma \tag{24}$$

*holds with a probability of at least $1 - \varepsilon$.*

*Let us now study two specific predictor functions in $\mathcal{F}$: An optimal predictor that minimizes the out-of-sample error $y^\star(\mathbf{x}) = \boldsymbol{\phi}^\top(\mathbf{x})\boldsymbol{\theta}^\star$, where $\boldsymbol{\theta}^\star = \mathbb{E}[\boldsymbol{\phi}(\mathbf{x})\boldsymbol{\phi}^\top(\mathbf{x})]^\dagger \mathbb{E}[\boldsymbol{\phi}(\mathbf{x})y]$ is a bounded vector. The learned predictor $y^\star(\mathbf{x}; \boldsymbol{\lambda}_n) = \boldsymbol{\phi}^\top(\mathbf{x})\boldsymbol{\theta}_n$, where $\boldsymbol{\theta}_n$ is a minimizer of (14). This vector is also bounded because the minimizer of (14) coincides with that of*

$$\underset{\boldsymbol{\theta}:\|\boldsymbol{\psi}\odot\boldsymbol{\theta}\|_1 \leq \gamma}{\arg\min} \|\mathbf{y} - \boldsymbol{\Phi}\boldsymbol{\theta}\|_2$$

*for some value of $0 \leq \gamma < \infty$. Thus both $\|\boldsymbol{\theta}^\star\|_1$ and $\|\boldsymbol{\theta}_n\|_1$ are bounded by some $P$ and (24) applies to the optimal and learned predictors, denoted by $y^\star$ and $\widehat{y}^n$ for brevity.*

*Since $\boldsymbol{\theta}_n$ minimizes the criterion in (14), it follows that*

$$\sqrt{R_n(\widehat{y}^n)} + n^{-1/2}\|\boldsymbol{\psi} \odot \boldsymbol{\theta}_n\|_1 \leq \sqrt{R_n(y^\star)} + n^{-1/2}\|\boldsymbol{\psi} \odot \boldsymbol{\theta}^\star\|_1, \quad \forall n$$

*After rearranging, we have*

$$\sqrt{R_n(\widehat{y}^n)} - \sqrt{R_n(y^\star)} \leq n^{-1/2}(\|\boldsymbol{\psi} \odot \boldsymbol{\theta}^\star\|_1 - \|\boldsymbol{\psi} \odot \boldsymbol{\theta}_n\|_1)$$
$$\leq n^{-1/2}\|\boldsymbol{\psi} \odot \boldsymbol{\theta}^\star\|_1 \qquad (25)$$
$$\leq n^{-1/2}\beta P,$$

*where $\beta = \|\boldsymbol{\psi}\|_\infty$. Multiplying both sides of the equality by the positive quantity $(\sqrt{R_n(\widehat{y}^n)} + \sqrt{R_n(y^\star)})$, we have*

$$R_n(\widehat{y}^n) - R_n(y^\star) \leq (\sqrt{R_n(\widehat{y}^n)} + \sqrt{R_n(y^\star)})n^{-1/2}\beta P$$
$$\leq (2\sqrt{R_n(y^\star)} + n^{-1/2}\beta P\,)n^{-1/2}\beta P, \qquad (26)$$

*where the second inequality follows from using (25). Finally, by definition $R(y^\star) \leq R(\widehat{y}^n)$ and we have that*

$$R(\widehat{y}^n) \leq R_n(\widehat{y}^n) + (P+1)^2\sigma$$
$$\leq R_n(y^\star) + (P+1)^2\sigma + (2\sqrt{R_n(y^\star)} + n^{-1/2}\beta P\,)n^{-1/2}\beta P$$
$$\leq R(y^\star) + 2(P+1)^2\sigma + (2\sqrt{R(y^\star) + (P+1)^2\sigma} + n^{-1/2}\beta P\,)n^{-1/2}\beta P \qquad (27)$$
$$= R(y^\star) + 2(P+1)^2 B\sqrt{2} \cdot \sqrt{\frac{1}{n}\ln\frac{2(d+1)^2}{\varepsilon}} + \mathcal{O}(n^{-3/4}),$$

*with a probability of at least $1-\varepsilon$, where (24) was used in the first and third inequality and (26) was used in the second inequality.*

### 4.3 Distributional robustness

In the previous section, we showed that the out-of-sample error of $\widehat{y}(\mathbf{x}; \boldsymbol{\lambda}_n)$ approaches the minimum achievable MSE at a root-$n$ rate. We will now see that this predictor also provides robustness against distributional uncertainty for finite $n$.

The feature vector $\boldsymbol{\phi}(\mathbf{x}) : \mathcal{X} \to \mathbb{R}^d$ for any predictor in $\mathcal{F}$ leads to a distribution of the random variables $(\boldsymbol{\phi}, y)$ which we denote $p(\boldsymbol{\phi}, y)$. The out-of-sample MSE can then be written

$$\mathbb{E}\left[\left(y_{n+1} - \widehat{y}(\mathbf{x}_{n+1})\right)^2\right] \equiv \mathbb{E}_p\left[(y_{n+1} - \boldsymbol{\phi}_{n+1}^\top\boldsymbol{\theta})^2\right], \qquad (28)$$

Using $n$ i.i.d. samples $(\boldsymbol{\phi}_i, y_i)$, we can define a predictor in $\mathcal{F}$ that minimizes the MSE under the *least favourable distribution* among all plausible distributions that are consistent with the data. Such a predictor is called 'distributionally robust', see, e.g., Duchi & Namkoong (2018). To formalize a set of plausible distributions, we first define the empirical distribution

$$p_n(\boldsymbol{\phi}, y) = \frac{1}{n}\sum_{i=1}^{n}\delta(\boldsymbol{\phi} - \boldsymbol{\phi}_i,\ y - y_i) \qquad (29)$$

Then we consider a set of distributions

$$\{p : D(p_n, p) \leq \epsilon_n\}, \qquad (30)$$

where $D(p_n, p)$ is some divergence measure. A distributionally robust predictor minimizes the MSE under the least-favourable distribution in the set (30), viz.

$$\max_{p\,:\,D(p_n,p)\leq\epsilon_n} \mathbb{E}_p\left[(y_{n+1} - \boldsymbol{\phi}_{n+1}^\top\boldsymbol{\theta})^2\right] \qquad (31)$$

Several different divergence measures $D(p_n, p)$ have been considered in the literature, including Kullback-Leibler divergence, chi-square divergence, and so on. One popular divergence measure is the Wasserstein distance (Blanchet et al., 2019), which is defined as

$$D(p_n, p) = \inf_\pi\ \mathbb{E}_\pi\left[c(\boldsymbol{\phi}, y,\ \boldsymbol{\phi}', y')\right], \qquad (32)$$

where $c(\boldsymbol{\phi}, y, \ \boldsymbol{\phi}', y')$ is a nonnegative cost function and $\pi$ is a joint distribution over $(\boldsymbol{\phi}, y, \ \boldsymbol{\phi}', y')$ whose marginals equal $p_n(\boldsymbol{\phi}, y)$ and $p(\boldsymbol{\phi}', y')$, respectively. Thus $D(p_n, p)$ can be interpreted as measuring the expected cost of moving probability mass from one distribution to the other.

**Theorem 3** *Suppose the observed features are standardized so that $\frac{1}{n}\sum_{i=1}^{n}\phi_k^2(\mathbf{x}_i) = 1$. Then $\widehat{y}(\mathbf{x}; \boldsymbol{\lambda}_n)$ corresponds to a predictor that minimizes the out-of-sample MSE under the least favourable distribution in the set $\{p : D(p_n, p) \leq n^{-2}\}$, defined by a Wasserstein distance (32) with a cost function*

$$c(\boldsymbol{\phi}, y, \ \boldsymbol{\phi}', y') = \begin{cases} \|\boldsymbol{\phi} - \boldsymbol{\phi}'\|_\infty^2 & y = y', \\ \infty & otherwise. \end{cases} \tag{33}$$

*Thus the predictor is robust against distributional uncertainties in the features $\boldsymbol{\phi}$, which may be high-dimensional. Note that the size of the distribution set shrinks with $n$.*

**Proof 3** *When the features are standardized, then $\boldsymbol{\psi} = \mathbf{1}$ and (14) becomes*

$$\arg\min_{\boldsymbol{\theta}} \sqrt{\frac{1}{n}\|\mathbf{y} - \boldsymbol{\Phi}\boldsymbol{\theta}\|_2^2} + \frac{1}{\sqrt{n}}\|\boldsymbol{\theta}\|_1. \tag{34}$$

*Using Theorem 1 in (Blanchet et al., 2019), (34) corresponds to a predictor that minimizes (31) with divergence bound $\epsilon_n = n^{-2}$.*

### 4.4 In-sample robustness

When learning $\widehat{y}(\mathbf{x}; \boldsymbol{\lambda}_n)$ it is possible that the observed covariates themselves are subject to errors so that the dataset is:

$$\widetilde{\mathcal{D}}_n = \{(\widetilde{\mathbf{x}}_1, y_1), \dots, (\widetilde{\mathbf{x}}_n, y_n)\}$$

Then the true feature vector $\boldsymbol{\phi}_i = \boldsymbol{\phi}(\mathbf{x}_i)$ can be viewed as a perturbed version of the observed vector $\widetilde{\boldsymbol{\phi}}_i = \boldsymbol{\phi}(\widetilde{\mathbf{x}}_i)$, where the perturbation $\boldsymbol{\delta}_i = \boldsymbol{\phi}_i - \widetilde{\boldsymbol{\phi}}_i$ is unknown. This problem (aka. errors-in-variables) leads to yet another interpretation of the predictor $\widehat{y}(\mathbf{x}; \boldsymbol{\lambda}_n)$.

**Theorem 4** *Consider the bounded set of possible in-sample perturbations:*

$$\mathcal{S}_n = \left\{ \boldsymbol{\delta}_1, \dots, \boldsymbol{\delta}_n \ : \ \mathbb{E}_{p_n}\left[\delta_k^2\right] \leq n^{-1}\,\mathbb{E}_{p_n}\left[\widetilde{\phi}_k^2\right], \ \forall k = 1, \dots, d \right\}$$

*Then $\widehat{y}(\mathbf{x}; \boldsymbol{\lambda}_n)$ corresponds to a predictor that minimizes the in-sample root-MSE under the least-favourable perturbations in $\mathcal{S}_n$:*

$$\max_{\{\boldsymbol{\delta}_i\}\in\mathcal{S}_n} \ \sqrt{\mathbb{E}_{p_n}\left[(y - (\widetilde{\boldsymbol{\phi}} + \boldsymbol{\delta})^\top\boldsymbol{\theta})^2\right]}, \tag{35}$$

*where $\widehat{y} = (\widetilde{\boldsymbol{\phi}} + \boldsymbol{\delta})^\top\boldsymbol{\theta} \in \mathcal{F}$.*

**Proof 4** *The problem in (35) can be written as:*

$$\max_{\{\boldsymbol{\delta}_i\}\in\mathcal{S}} \frac{1}{\sqrt{n}}\|\mathbf{y} - (\widetilde{\boldsymbol{\Phi}} + \boldsymbol{\Delta})\boldsymbol{\theta}\|_2, \ \ where \ \boldsymbol{\Delta} \ = \ \begin{bmatrix} \boldsymbol{\delta}_1^\top \\ \vdots \\ \boldsymbol{\delta}_n^\top \end{bmatrix} \tag{36}$$

*Let $[\boldsymbol{\Delta}]_k$ denote the $k^{th}$ column of the matrix $\boldsymbol{\Delta}$. We can then upper bound the error as*

$$\max_{\{\boldsymbol{\delta}_i\}\in\mathcal{S}} \frac{1}{\sqrt{n}} \left\| \mathbf{y} - \widetilde{\boldsymbol{\Phi}}\boldsymbol{\theta} - \sum_{k=1}^{d}[\boldsymbol{\Delta}]_k\theta_k \right\|_2 \leq \max_{\{\boldsymbol{\delta}_i\}\in\mathcal{S}} \frac{1}{\sqrt{n}}\|\mathbf{y} - \widetilde{\boldsymbol{\Phi}}\boldsymbol{\theta}\|_2 + \frac{1}{\sqrt{n}}\sum_{k=1}^{d}\|[\boldsymbol{\Delta}]_k\theta_k\|_2,$$

$$\leq \frac{1}{\sqrt{n}}\|\mathbf{y} - \widetilde{\boldsymbol{\Phi}}\boldsymbol{\theta}\|_2 + \max_{\{\boldsymbol{\delta}_i\}\in\mathcal{S}} \frac{1}{\sqrt{n}}\sum_{k=1}^{d}\|[\boldsymbol{\Delta}]_k\|_2|\theta_k|, \tag{37}$$

$$\leq \frac{1}{\sqrt{n}}\|\mathbf{y} - \widetilde{\boldsymbol{\Phi}}\boldsymbol{\theta}\|_2 + \frac{1}{\sqrt{n}}\sum_{k=1}^{d}\sqrt{\mathbb{E}_{p_n}[\widetilde{\phi}_k^2]}|\theta_k|.$$

*where the bound is attainable when*

$$[\boldsymbol{\Delta}]_k = \sqrt{\mathbb{E}_{p_n}[\widetilde{\phi_k^2}]} \frac{\mathbf{y} - \widetilde{\boldsymbol{\Phi}}\boldsymbol{\theta}}{||\mathbf{y} - \widetilde{\boldsymbol{\Phi}}\boldsymbol{\theta}||_2}.$$

*But the bound is of the same form as the cost function in (14). Thus solving problem (14) implies the mini-mization of (36). Theorem 1 in Xu et al. (2009) established a connection between a square-root maximization problem in the form of (36) and the penalized form on the right-hand side of (37).*

## 5 Numerical Experiment

In the previous sections we have showed several computational and theoretical properties of the predictor function $\widehat{y}(\mathbf{x}; \boldsymbol{\lambda}_n)$ which we shall call the SPICE-predictor. In this section we present a numerical experiment for sake of illustration.

### 5.1 Setup

We observe a stream of $n$ samples generated by the following (unknown) process

$$\begin{aligned} \mathbf{x} &\sim \text{Uniform}([0, 10]^2), \\ y|\mathbf{x} &\sim \text{GP}(0, k(\mathbf{x}, \mathbf{x}') + \sigma^2 \delta(\mathbf{x}, \mathbf{x}')), \end{aligned} \tag{38}$$

where

$$k(\mathbf{x}, \mathbf{x}') = \sigma^2 \left(1 + \frac{\sqrt{3}}{l}||\mathbf{x} - \mathbf{x}'||_2\right) \exp\left(-\frac{\sqrt{3}}{l}||\mathbf{x} - \mathbf{x}'||_2\right).$$

with noise variance $\sigma = 2$ and scale $l = 7$. In other words, $\mathbf{x}$ is a two-dimensional covariate drawn from a uniform distribution and $y$ is drawn from a Gaussian process (GP) with zero mean and a Matérn covariance function. A realization of the process above and $n$ training data points are shown in Figures 1a and 1e.

We consider a class $\mathcal{F}$ with $d = 100$ periodic feaatures $\{\phi_k(\mathbf{x})\}$ using the Laplacian eigenfunction basis (Solin & Särkkä, 2020). Note that this corresponds to a misspecified covariance model (4). We are interested in the online learning of predictors in $\mathcal{F}$, and use the least-squares (LS) and ridge regression methods as baseline references. Both methods can be implemented in an online fashion, but the latter requires fixing a regularization parameter. Here we simply set this parameter to 0.1 based on visual inspection.

### 5.2 Out-of-sample performance

For illustration, consider the predictions produced by the LS, ridge regression and SPICE methods, see Figure 1. As expected, the LS provides poor results at these sample sizes. Ridge regression with a fixed regularization parameter and SPICE with adaptively learned parameters appear to perform similarly here. To evaluate their out-of-sample errors, we compare the MSE against that of the oracle predictor based on the (oracle) Gaussian process repression (GPR) predictor in (38). Table 1 shows that the out-of-sample error of SPICE is lower than that of LS and ridge regression, and that the chosen class $\mathcal{F}$ is capable of predicting the GP in (38) well.

Following the discussion of effective degrees of freedom $df_n$ in Ruppert et al. (2003), we also provide a comparison between LS, SPICE and the oracle GPR predictors in Figure 2. While LS attains the maximum $df_n$ at $n = 100$, SPICE moderates its growth rate in a data-adaptive and online manner. The degrees of freedom of the oracle predictor increases gracefully and remains below its maximum value, even when $n$ increases beyond $d$.

### 5.3 Run-time

We report the runtimes of RIDGE (since LS is virtually identical) and SPICE, and consider a well-specified GPR predictor with covariance parameters learned using the maximum likelihood method as reference. As

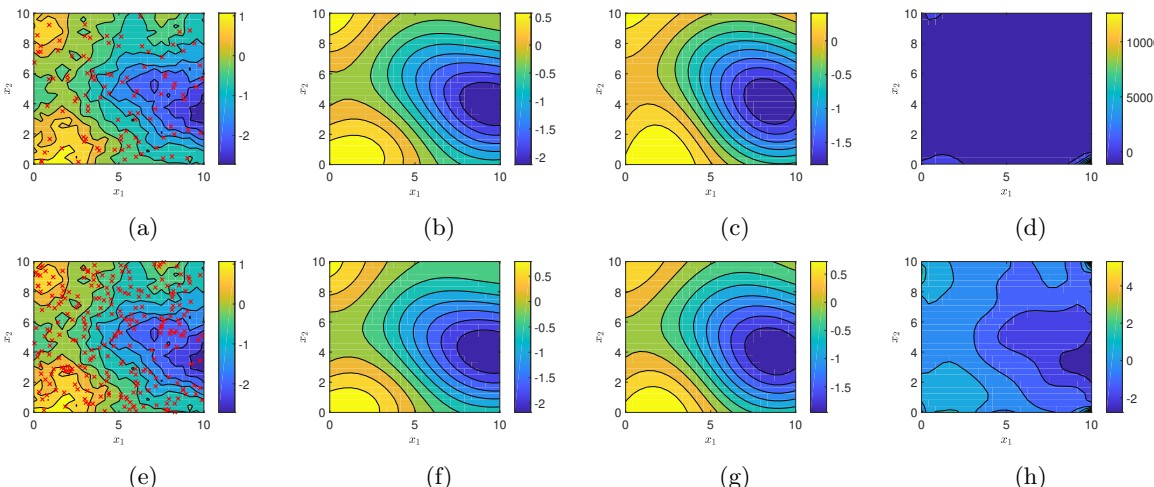

Figure 1: Contour plots. First column shows a realization of $y$ in (38) along with sampling patterns $\{\mathbf{x}_i\}_{i=1}^n \in \mathcal{X}$ for $n = 100$ and $n = 250$ (top and bottom rows, respectively). Second, third and fourth columns show the contour plots of the SPICE-predictor, ridge regression and the LS-predictor, respectively. All three predictors belong to $\mathcal{F}$.

| | MSE/MSE* | | |
|---|---|---|---|
| $n$ | LS | RIDGE | SPICE |
| 50 | $4.38 \times 10^4$ | 1.71 | 1.11 |
| 100 | 21.12 | 1.47 | 1.09 |
| 250 | 1.47 | 1.19 | 1.06 |
| 500 | 1.11 | 1.06 | 1.02 |

Table 1: Mean-square error (MSE) for LS and SPICE methods, normalized by MSE* of an oracle predictor which is given the unknown covariance function in (38). For a given set of training data $\mathcal{D}_n$, we compute the averaged squared error over 250 test points. The mean of this error is the MSE and was approximated using 100 different realizations of $\mathcal{D}_n$.

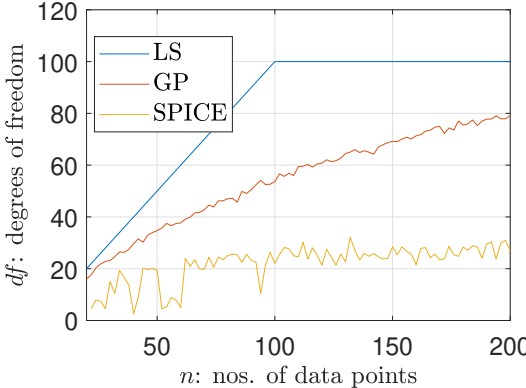

Figure 2: Plot of degrees of freedom $df_n$ against number of data points $n$ for LS, SPICE and oracle GP predictors.

can be seen in Table 2, the computational complexity of GPR is considerably higher than that of the online alternatives. RIDGE has a lower run-time than SPICE but both perform as $\mathcal{O}(n)$. A visualization of the trends for $0 < n \le 500$ is given in Figure 3.

| | Run times (ms) | | |
|---|---|---|---|
| $n$ | GPR | RIDGE | SPICE |
| 50 | 59 | 0.15 | 6.0 |
| 100 | 113.8 | 0.16 | 11.6 |
| 250 | 231.3 | 0.20 | 29.2 |
| 500 | 1507.2 | 0.22 | 58.6 |

Table 2: Run times for oracle GPR, RIDGE and SPICE. Note that the run times are computed for a single realization.

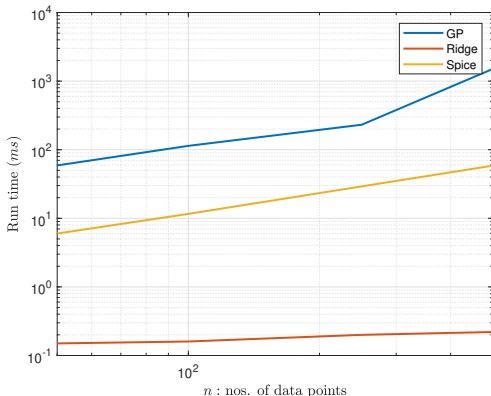

Figure 3: Plot of run time of a single run for learning parameters and prediction against versus of data points $n$ for RIDGE, SPICE and oracle GP predictors.

### 5.4 Robustness

In Theorems 3 and 4 established two different types of robustness results with respect to the features $\phi(\mathbf{x})$. We therefore study the predictive performance when the training and test distributions over $(\phi, y)$ diverge.

Specifically, the covariates in the training data are drawn as $\widetilde{\mathbf{x}}|\mathbf{x} \sim \mathcal{N}(\mathbf{x}, \sigma_x^2 \mathbf{I})$, where $\mathbf{x}$ follows (38). This results in a distribution shift over the features (aka. errors-in-variables). Figure 4 evaluates the test MSE when data drawn from (38). We see that the out-of-sample error for RIDGE and SPICE increases consistently with $\sigma_x$, while LS produces very poor and inconsistent results. SPICE is notably more robust against this distributional shift and these results corroborated the derived robustness properties.

## 6 Conclusion

We considered the problem of learning model-based linear smoothers online. If the model parameters were fixed, the resulting predictor – which includes Gaussian process regression and kriging methods – can readily be computed sequentially. Since the model parameters unknown, however, they must be learned from data, using, e.g., maximum likelihood or cross-validation methods. But implementing them when the data arrives as a stream requires either recomputing the predictor which is computationally prohibitive or resorting to approximations. In either case, these approaches do not offer clear-cut results on the statistical properties of the resulting predictor.

We applied a covariance-fitting method to learn the model parameters, which was initially developed for spectral estimation. We first used its computational properties to show that the resulting predictor can be computed sequentially. We then derived finite out-of-sample performance guarantees of the resulting predictor and showed that its error approaches the minimum achievable level at root-$n$ rate. Finally, we established connections to the distributional robustness literature by showing that the predictor is robust against distri-

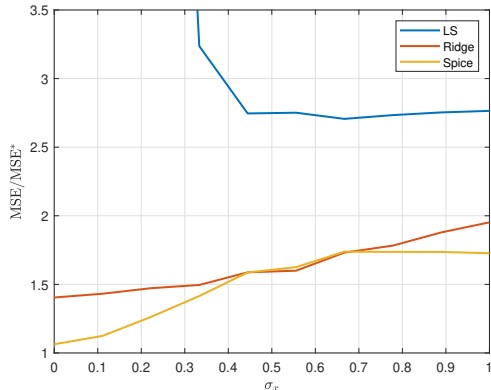

Figure 4: Out-of-sample error when the distribution of training data diverges from that of the test data. MSE (normalized by minimal MSE*) versus perturbation noise level $\sigma_x$ on training data ($n = 100$). Evaluation based on 500 Monte Carlo runs.

butional uncertainties and errors in the covariate training data. The performance, computational complexity and robustness of the proposed method were illustrated in a numerical experiment.

**Acknowledgments** We thank the anonymous reviewers from TMLR for their constructive feedback. This research was partially supported by the Swedish Research Council under contracts 2018-05040 and 2021-05022, Wallenberg AI, Autonomous Systems and Software Program (WASP) funded by Knut and Alice Wallenberg Foundation, and by Kjell och Märta Beijer Foundation.

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

# A  Appendix: Sequential computation

Here we provide a pseudocode for computing $\widehat{y}(\mathbf{x}; \boldsymbol{\lambda}_n) \equiv \boldsymbol{\phi}^\top(\mathbf{x})\boldsymbol{\theta}_n$ sequentially. This is accomplished via a cyclic minimization of the convex problem (14).

First, define the $k$th column as $\mathbf{c}_k = [\boldsymbol{\Phi}]_k$ and $\widetilde{\mathbf{y}}_k = \mathbf{y} - \sum_{j \neq k} \mathbf{c}_j \theta_j$. Then the cost function in (14) can be equivalently expressed as

$$V(\theta_k) = (\|\widetilde{\mathbf{y}}_k - \mathbf{c}_k \theta_k\|_2^2)^{1/2} + \psi_k |\theta_k| + C_k, \tag{39}$$

and minimized cyclically, one coordinate $k = 1, 2, \ldots, d$ at a time. It was shown in (Zachariah & Stoica, 2015) that the minimizer of (39) is given by

$$\widehat{\theta}_k = \begin{cases} s_k r_k, & \text{if } \sqrt{n-1}\gamma_k > \sqrt{\alpha_k \beta_k - \gamma_k^2} \\ 0, & \text{else} \end{cases} \tag{40}$$

where

$$\alpha_k = \|\widetilde{\mathbf{y}}_k\|_2^2, \quad \beta_k = \|\mathbf{c}_k\|_2^2, \quad \gamma_k = |\mathbf{c}_k^\top \widetilde{\mathbf{y}}_k| \tag{41}$$

and

$$s_k = \text{sign}(\mathbf{c}_k^\top \widetilde{\mathbf{y}}_k), \quad r_k = \frac{\gamma_k}{\beta_k} - \frac{1}{\beta_k}\left(\frac{\alpha_k \beta_k - \gamma_k^2}{n-1}\right) \tag{42}$$

The key observation here is that the variables (41) are all expressible using the recursively computable quantities in (15) (along with a prior parameter iterate $\boldsymbol{\theta}'$ which is initialized at $\mathbf{0}$).

To arrive at this conclusion, define the following global variables

$$\mathbf{z} = \mathbf{y} - \boldsymbol{\Phi}\boldsymbol{\theta}', \quad g_0 = \|\mathbf{z}\|_2^2, \quad \mathbf{g} = \boldsymbol{\Phi}^\top \mathbf{z} \tag{43}$$

which can be expressed using (15). Then we have that $\widetilde{\mathbf{y}}_k \equiv \mathbf{z} + \mathbf{c}_k \theta'_k$, so that we can express (41) in terms of the global variables:

$$\alpha_k = g_0 + A^n_{kk}\theta_k^2 + 2g_k\theta'_k, \quad \beta_k = A^n_{kk}, \quad \gamma_k = |g_k + A^n_{kk}\theta'_k|, \quad s_k = \text{sign}(g_k + A^n_{kk}\theta'_k) \tag{44}$$

Finally we can express the cyclic minimization approach in recursive form as outlined in Algorithm 1, initializing $\boldsymbol{\theta}' = \mathbf{0}$ at $n = 1$. Additional computational considerations are developed in Sec. III of (Zachariah & Stoica, 2015). Link to implementation is provided here: `https://github.com/Muhammad-Osama`.

---

**Algorithm 1** : SPICE-predictor

---

1: Input: $(\mathbf{x}_n, y_n)$ and $\mathbf{x}$ (and initial $\boldsymbol{\theta}'$)
2: Update recursive variables $\mathbf{A}^n, \mathbf{b}^n$ and $c^n$ in (15)
3: Set global variables $g_0 = c^n + \boldsymbol{\theta}'^\top \mathbf{A}^n \boldsymbol{\theta}' - 2\boldsymbol{\theta}'^\top \mathbf{b}^n$ and $\mathbf{g} = \mathbf{b}^n - \mathbf{A}^n \boldsymbol{\theta}'$
4: **repeat**
5: $\quad k = 1, \ldots, d$
6: $\quad$ Compute $\alpha_k, \beta_k, \gamma_k$ and $s_k$ in (44)
7: $\quad$ Compute $\widehat{\theta}_k$ in (40)
8: $\quad$ Update global variables $g_0 := g_0 + A^n_{kk}(\theta'_k - \widehat{\theta}_k)^2 + 2(\theta'_k - \widehat{\theta}_k)g_k$ and $\mathbf{g} := \mathbf{g} + [\mathbf{A}^n]_k(\theta'_k - \widehat{\theta}_k)$
9: $\quad$ Update $\theta'_k := \widehat{\theta}_k$
10: **until** number of iterations equals $L$
11: Output: $\widehat{y}(\mathbf{x}) = \boldsymbol{\phi}^\top(\mathbf{x})\widehat{\boldsymbol{\theta}}$

---

