# OpenReview forum: "Online Learning for Prediction via Covariance Fitting: Computation, Performance and Robustness"
_TMLR — Accepted by TMLR_

### Review · Reviewer_ZrgT · 2022-09-20

**Summary Of Contributions:**

The submission considers a linear smoother predictor for streaming data. Specifically, data $(x_i, y_i)$ comes in an online fashion, and the learner wants to learn/update the prediction of an input $\hat{y}(x)$ as observing a new sample without learning from the scratch (i.e., iterating again over all previous samples). The proposed estimator fits the model covariance matrix $C_\lambda$ using the criterion known as SPICE and plug it to construct the predictor. Generalization and perturbation-robustness guarantees are given for the overall prediction performance.

As far as I can tell, not much thing is new in this paper: the algorithm is already some well-known one in literature, and the analysis is seemingly an straight-forward application of standard techniques.


**Broader Impact Concerns:**

No specific broader impact concerns.

**Requested Changes:**

The paper could have been more interesting if you also study non-i.i.d. streaming setup. For example, some data could have been corrupted (not just perturbed in a small amount), or underlying distribution can slowly change, etc. In its current setup, I do not see any particular challenge by considering an online setup.

**Strengths And Weaknesses:**

Strength

- Perturbation-robustness guarantees in both test and training times are studied.

- Theoretical findings are demonstrated via numerical experiments.

Weakness

- The main reason that the update can be performed sequentially is because the sufficient statistics needed for solving the least square problem can be expressed as summations over samples. Given this, I do not think that it is surprising to expect the predictor can be updated in an online manner.

- Generalization bounds given in the paper are standard applications of basic probabilistic inequalities and has nothing to do with online aspect of the problem.

---

> ### Author Response · Authors · 2022-09-22
> **Reply to Reviewer ZrgT**
>
>
> We thank the reviewer for the comments and take the opportunity to clarify some critical points of the paper.
>
> > \#1: ``The main reason that the update can be performed sequentially is because the sufficient statistics needed for solving the least square problem can be expressed as summations over samples. Given this, I do not think that it is surprising to expect the predictor can be updated in an online manner.''
>
> For clarity, let us first restate the main contribution of our paper: to provide a method that learns a predictor $\hat{y}(x; \lambda_n)$ in (i) a sequential manner with (ii) distribution-free and finite out-of-sample performance guarantees.
>
> *Neither* cross validation $\hat{y}(x; \lambda^{\text{CV}}_n)$,  maximum likelihood $\hat{y}(x; \lambda^{\text{ML}}_n)$, nor any other parameter tuning method we are aware of, enable (i) or provide (ii).
>
> The fact that the covariance-fitting criterion leads to a regularized problem with `sufficient statistics' is not some deficiency but a feature: it is precisely what enables (i) *unlike* the cited alternatives, and *in addition* we provide (ii) for the first time. (Compare for instance the cited books: Ruppert et al. 2003, ch. 5.2-3; Bishop, 2006, ch. 1.3;  Rasmussen \& Williams, 2006, ch. 5.4; Hastie et al., 2008, ch. 7.10; Stein, 2012, ch. 6.8.)
>
> We will make these points more explicit in a revised manuscript.
>
>
> > \#2: ``Generalization bounds given in the paper are standard applications of basic probabilistic inequalities and has nothing to do with online aspect of the problem.''
>
> The fact that we used well-established tools to derive new generalization bounds is not some deficiency but a virtue: it shows that  out-of-sample guarantees of parameter tuning via covariance-fitting are derivable without recourse into nonstandard theory.
>
> The finite-sample result is indeed general, but applicable to the online -- that is, data streaming -- aspect of the problem in so far as it applies at any given length $n$ of the stream.
>
>
> > \#3: ``In its current setup, I do not see any particular challenge by considering an online setup.''
>
> The particular challenge is that *neither* cross validation, maximum likelihood, nor any other parameter tuning method we are aware of, enable sequential computation of $\hat{y}(x; \lambda_n)$  nor provide distribution-free and finite out-of-sample guarantees for its performance.
>
>
> > \#4: ``The paper could have been more interesting if you also study non-i.i.d. streaming setup. For example, some data could have been corrupted (not just perturbed in a small amount), or underlying distribution can slowly change, etc.''
>
> Non-i.i.d. streaming (adversarial or not) is an extremely wide class of problems, well beyond the scope of our paper or even the use of standard tools of analysis.
>
>
> > \#5: ``As far as I can tell, not much thing is new in this paper: the algorithm is already some well-known one in literature, and the analysis is seemingly an straight-forward application of standard techniques.''
>
> The covariance-fitting criterion was indeed, as we write, ``first proposed in the context of spectral estimation'', and further developed in the cited references. Our paper is the *first to derive its finite-sample properties when applied to prediction*.
>
> To the best of our knowledge, no corresponding result exists for cross-validation, maximum likelihood or any alternate techniques for fitting $\lambda_n$. The fact that this analysis was amenable by the use of `standard techniques' is in our view a virtue, not a vice.

---

### Review · Reviewer_cTRW · 2022-09-28

**Summary Of Contributions:**

Summary of contributions： This paper proposes a covariance-fitting method to solve an online learning problem together with three types of theoretical guarantees, including computational complexity, excess risk bound, and robustness. Specifically, the proposed online learning method can be updated sequentially, which is more computationally efficient than existing techniques. Moreover, its out-of-sample error approaches the minimum achievable level at a root-n rate, where n is the number of data samples. Finally, it minimizes the out-of-sample error under the least favorable distribution within a given Wasserstein ball from the empirical distribution. It is also robust against errors in the covariate training data.



**Broader Impact Concerns:**

There is no such concern for this theoretical paper.

**Requested Changes:**

See above

**Strengths And Weaknesses:**


Strengths: The theoretical results and proofs are presented in a clear way, and the setting considered in the paper is general enough to be interested in the audience of TMLR.

Weaknesses and Requested changes:

1. This paper lacks a more detailed discussion on related work. Such an online prediction problem should have been studied thoroughly in recent years, but most references in the current paper are before 2010.
2. Lack of experiment justification: This paper advocates the proposed algorithm from three aspects, computational efficiency, out-of-sample generalization, and robustness. However, the experimental section only compares the out-of-sample error with simple baseline methods, including LS and ridge regression. More empirical results should be provided to demonstrate the proposed method's computational efficiency (run time comparison) and robustness (corresponding to the two notions considered in the paper).

Minor comments:

1. Typo 3rd line after equation (2): “This is includes”
2. Last line on page 3, “is provided at TOAPPEAR”
3. The paper can be more self-contained by adding remarks to discuss the results used from Zachariah & Stoica (2015), Blanchet et al. (2019), and Xu et al. (2009).

---

### Review · Reviewer_tGLV · 2022-11-01

**Summary Of Contributions:**

This paper studies the regression problem where the output values of the sampling model have conditional covariance, conditioned on input vector. The covariance also depends on a training parameter (which confuses me a bit). The hypothesis space is designed to be all the weighted sums of training labels, with weights being functions of covariates. A convex optimization-based online algorithm is proposed and analyzed. The theoretical analysis provides computational complexity bound, out-of-sample error bound, and robustness analysis.

**Broader Impact Concerns:**

This paper studies a machine learning algorithm. I have not any ethical concerns.

**Requested Changes:**

1. I feel confused about the claim (in the abstract) that MLE cannot be made online. I think that is easy. For example, just a google search gives me the following papers:

* Surace, S.C. and Pfister, J.P., 2018. Online maximum-likelihood estimation of the parameters of partially observed diffusion processes. IEEE transactions on automatic control, 64(7), pp.2814-2829.

* Rohde, D. and Cappé, O., 2011, June. Online maximum-likelihood estimation for latent factor models. In 2011 IEEE Statistical Signal Processing Workshop (SSP) (pp. 565-568). IEEE.

* Malik, S. and Enzner, G., 2010, March. Online maximum-likelihood learning of time-varying dynamical models in block-frequency-domain. In 2010 IEEE International Conference on Acoustics, Speech and Signal Processing (pp. 3822-3825). IEEE.

I do not mean to ask the authors to cite them, but just that the above claim is confusing.

2. I am not certain about the claim at the end of page 2 that "using data-dependent parameters λ in (4) does not readily provide any out-of-sample prediction performance guarantees." In particular, we see some works in the literature, and such data-dependent approaches are referred to as "adaptive algorithms". For example,

* Caponnetto, Andrea; Yao, Yuan. Cross-validation-based adaptation for regularization operators in learning theory. Anal. Appl. (Singap.) 8 (2010), no. 2, 161--183.

* Cai, T. Tony; Yuan, Ming. Minimax and adaptive prediction for functional linear regression. J. Amer. Statist. Assoc. 107 (2012), no. 499, 1201--1216.

3. The norm notation in (8) is not defined.

4. The method of computing the features in (3) is not specified. For this reason, some part of the proof of Theorem 1 needs to be re-assessed after the revision of the paper.

5. I double the sentence "It can also be shown that the minimizer of the concentrated criterion V(θ(λ),λ) equals cλ◦, that is a rescaling of the covariance-based parameters (8)" between Equations (10) and (11). In particular, I think such equivalence may fail when the norm of the vector y does not equal 1. I suggest that the authors provide more details on this point.

6. In (12), the factor 1/sqrt(n) of the second term on the right-hand side should be sqrt(n).

7. Please cite the specific results in Zachariah & Stoica (2015) for the claimed computational complexity.

8. Link "toappear" in page 3 is broken.

9. I feel confused about (3), which claims that the data is not iid. Is there any target function that the training process is planned to recover? Is the vector lambda unknown? Is there any mechanism (for example, procedure, or probability distribution) on data generation? Is there any assumption on the input space script-X? For example, is script-X a metric space? Is it a topological space? Or is it just a compact subset of some Euclidean space? Is (2) an assumption, or the consequence of some other assumptions?

10. Please provide details on the iteration procedure.

11. Right after (13), we see the description of "an unknown distribution p(x, y)". Is this distribution also used to generate the training data Dn?

12. Right after (15), the authors mentioned the probability "at least 1 − ε". What is the associated probability distribution? Note that we already see means in (15), so both sides of the inequality should already be constants.

13. Right after (17), a new assumption that "φ(x)" is bounded, is introduced. Please include this assumption in the theorem.

14. For the paragraph before (22): how is the "optimal predictor y*"  defined? Please provide more details to explain the claim "Since z and Σ are bounded, the parameters are also bounded such that k...... ≤ P for some P ." Note that the boundedness of a minimization problem coefficient does not guarantee the boundedness of its solution. For example, min(1/x + ax) over (0,inf) leads to unbounded solution x*=1/sqrt(a) when the constant "a" takes values in a bounded set (0,1).

**Strengths And Weaknesses:**

Strengths:

1. The research problem is interesting.
2. Theoretical analysis, if technically correct, would be very good contributions.

Weaknesses:

1. The proposed iteration procedures appear to be a straightforward application of weighted square-root LASSO. This raises concerns about the novelty of the paper.
2. There are undefined notations. We find assumptions introduced in the proofs without being claimed in the statement of the theorem.
3. Some technical details are missing, for example, there is no description of the main proposed iteration algorithm.
4. Some writing problems as specified below.

---

> ### Author Response · Authors · 2022-11-09
> **Our reply, part a)**
>
> We thank the reviewer for the detailed feedback and thoughtful questions. We are confident that by addressing them in the revised manuscript, its readability will improve substantially.
>
> >\#1: ``The proposed iteration procedures appear to be a straightforward application of weighted square-root LASSO. This raises concerns about the novelty of the paper.''
>
> The novelty of the paper is to apply the SPICE methodology for fitting the covariance parameters for (i) online prediction with (ii) out-of-sample guarantees. To this end, we do indeed leverage a connection between the SPICE criterion and a particular form of weighted square-root LASSO that was established in the cited signal processing literature. We will restate these points more clearly in the revision.
>
> > \#2: ``There are undefined notations. We find assumptions introduced in the proofs without being claimed in the statement of the theorem.''
>
> We agree with the reviewer and will revise the manuscript accordingly (see replies below).
>
> > \#3: ``Some technical details are missing, for example, there is no description of the main proposed iteration algorithm"
>
> For completeness, we have decided to include the iterative algorithm in the appendix of the paper.
>
> > \#4.1: ``I feel confused about the claim (in the abstract) that MLE cannot be made online. [...] I do not mean to ask the authors to cite [the papers], but just that the above claim is confusing..''
>
> The abstract stated that neither cross-validation nor maximum likelihood estimation are suitable for covariance-based when training data arrives in a streaming fashion. In light of the reviewer's comments, we believe this statement can easily be misread.
>
> It is of course possible to update the model parameters online by gradient steps to *approximate* the maximum likelihood $\lambda^{\text{ML}}_{n}$ for any $n$. But (i) the weights given by (5) will have to be recomputed -- including the inverse of covariance matrix (6) -- for *every* new data point. While this can also be done approximately, to the best of our knowledge this approach does not (ii) offer any performance guarantees of the resulting approximated predictor. The SPICE methodology enables us to address both (i) and (ii).
>
> We will revise the statement in the abstract and make the claims more precise throughout the paper.
>
> > \#4.2: ``I am not certain about the claim at the end of page 2 that "using data-dependent parameters $\lambda$ in (4) does not readily provide any out-of-sample prediction performance guarantees." In particular, we see some works in the literature, and such data-dependent approaches are referred to as "adaptive algorithms". [Cf. Caronetto \& Yao, 2010; Cai \& Yuan, 2012.]
>
> We thank the reviewer for the references to these `adaptive' methods and we find it useful to cite (Cai \& Yuan, 2012) which do study data-dependent hyperparameters.
>
> Theorem 4 in that paper provides a useful contrast with the type of out-of-sample guarantee that we consider: ours is non-asymptotic and applicable to the online learning setting.
>
> > \#4.3: ``The norm notation in (8) is not defined."
>
> We will add the definition.
>
> > \#4.4: ``The method of computing the features in (3) is not specified. For this reason, some part of the proof of Theorem 1 needs to be re-assessed after the revision of the paper."
>
> We will clarify this point in the revised manuscript: In principle, we can use any bounded feature function $\phi(x)$. The cited papers following eq. (3) recommend the use of Fourier-type features due to their excellent covariance approximating properties. Indeed, this is the type of $\phi$ we consider in our numerical experiments.
>
> > \#4.5: ``I double the sentence "It can also be shown that the minimizer of the concentrated criterion $V(\theta(\lambda),\lambda)$ equals cλ◦, that is a rescaling of the covariance-based parameters (8)" between Equations (10) and (11). In particular, I think such equivalence may fail when the norm of the vector y does not equal 1. I suggest that the authors provide more details on this point."
> \end{quote}
>
> For completeness of the proof, we will include a part that proves why the equivalence holds (for any $||y ||_2 >0$).

---

> > ### Author Response · Authors · 2022-11-09
> > **Our reply, part b)**
> >
> >
> > > \#4.6: ``In (12), the factor 1/sqrt(n) of the second term on the right-hand side should be sqrt(n)."
> >
> > By inserting (11) into (10) we obtain $2\sqrt{n}$ $|| y - \Phi \theta ||_2$ + $2\sqrt{n}$ $\sum^d_{k=1} |\psi_k|\theta_k|$.
> >
> > Dividing this expression by $2n$, enables us to express the problem in the form of (12) which therefore was correct and is subsequently used in the main result.
> >
> > > \#4.7 ``Please cite the specific results in Zachariah \& Stoica (2015) for the claimed computational complexity."
> >
> > We will update this citation.
> >
> > > \#4.8 ``Link "toappear" in page 3 is broken."
> >
> > We deliberately removed this link to comply with the double-blind review process.
> >
> > > \#4.9a ``I feel confused about (3), which claims that the data is not iid.
> >
> > We first consider predictors (4) based on a conditional covariance model (3) (aka. Gaussian process regression model/kriging). We then evaluate the predictive performance of a fitted predictor (using $\hat{\lambda}(\mathcal{D}_n)$) when the data pairs $(x_i, y_i)$ are drawn i.i.d. We shall make this point explicit.
> >
> > > \#4.9b ``Is there any target function that the training process is planned to recover? Is the vector lambda unknown?"
> >
> > In the covariance model (3), the nonnegative vector $\lambda$ is an unknown model parameter with a corresponding predictor $\hat{y}(x; \lambda)$. Therefore any training process corresponds to fitting $\lambda$ with $\mathcal{D}_n$ in some way -- ML, CV and SPICE being different examples. All parameterized predictors $\hat{y}(x; \lambda)$ belong to the class $\mathcal{F}$ (14) and we are asking how closely a fitted $\hat{y}(x; \hat{\lambda}_n)$ can come to the optimal out-of-sample performance in $\mathcal{F}$? The answer we provide is in Theorem 2.2.
> >
> > We will clarify these points in the revision.
> >
> > > \#4.9c ``Is there any mechanism (for example, procedure, or probability distribution) on data generation?"
> >
> > Yes, an unknown data-generating process where each pair $(x_i,y_i)$ is drawn i.i.d. Note that the distribution is not necessarily included in the distributional models parameterized by the covariance model. In other words, we do not assume that the covariance model is well-specified.
> >
> > > \#4.9d ``Is there any assumption on the input space script-X? For example, is script-X a metric space? Is it a topological space? Or is it just a compact subset of some Euclidean space? Is (2) an assumption, or the consequence of some other assumptions?"
> >
> > There is no assumption about the structure of $\mathcal{X}$ per se, rather we assume that the feature vector $\phi(x)$ forms a bounded random variable in $\mathbb{R}^d$. We will clarify this point.
> >
> > > \#4.10 ``Please provide details on the iteration procedure."
> >
> > The iterative algorithm will now be provided in the appendix.
> >
> > > \#4.11 ``Right after (13), we see the description of "an unknown distribution p(x, y)". Is this distribution also used to generate the training data Dn?."
> >
> > Yes, we will make this point clearer and more explicit: To study the out-of-sample performance of the predictor, we will consider a stream of i.i.d. samples $(x_1,y_1), \dots, (x_n,y_n), (x_{n+1},y_{n+1})$, drawn from an unknown distribution
> > where the $n$ first samples are the training data used for prediction.

---

> > > ### Author Response · Authors · 2022-11-09
> > > **Our reply, part c)**
> > >
> > > > \#4.12 ``Right after (15), the authors mentioned the probability "at least $1-\varepsilon$". What is the associated probability distribution? Note that we already see means in (15), so both sides of the inequality should already be constants."
> > >
> > > The expectations in (15) are taken over the (next) sample $(x_{n+1}, y_{n+1})$. The fitted predictor $\hat{y}(x; \lambda^\circ)$ on the LHS, however, is also a function of the training data so that its MSE depends on the realization of $\mathcal{D}_n$. The inequality therefore holds with a probability $1-\varepsilon$ with the associated distribution is over $\mathcal{D}_n$.
> > >
> > > We will revise the manuscript, especially the setup for the theorem, to make the result more explicit.
> > >
> > > > \#4.13 ``Right after (17), a new assumption that "$\phi(x)$" is bounded, is introduced. Please include this assumption in the theorem.."
> > >
> > > We will do so.
> > >
> > > > \#4.14 ``For the paragraph before (22): how is the "optimal predictor $y*$" defined? Please provide more details to explain the claim "Since $z$ and $\Sigma$ are bounded, the parameters are also bounded such that $k...... \leq P$ for some $P$." Note that the boundedness of a minimization problem coefficient does not guarantee the boundedness of its solution."
> > >
> > > It is readily shown that the MSE-optimal predictor is $y^*(x) = \phi^\top(x) \theta^*$, where $\theta^* =  \mathbb{E}[\phi(x) \phi^\top(x)]^\dagger \mathbb{E}[\phi(x) y]$ is bounded. We will explicate this in the proof.
> > >
> > > Now consider $\theta^\circ$ that minimizes the convex penalized function $|| y - \Phi \theta ||_2 +  ||\psi \odot \theta||_1$, where $y$, $\Phi$ and $\psi >0$ are bounded variables.
> > >
> > > Then there exists some $0\leq \gamma < \infty$ for which the constrained convex minimization problem $\min_{\theta : || \psi \odot \theta ||_1 \leq \gamma} || y - \Phi \theta ||_2$ has the same minimizer $\theta^\circ$ as for the convex penalized function. Therefore this parameter vector is bounded: $\theta^\circ \in$ { $\theta :  || \psi \odot \theta ||_1 \leq \gamma$ }.
> > >
> > > PS. We anticipate the revised manuscript to be uploaded early next week.

---

> > > > ### Comment · Reviewer_tGLV · 2022-11-14
> > > > **Thanks for the detailed responses**
> > > >
> > > > We appreciate the detailed responses and we look forward to studying the revised manuscript.

---

> > > > > ### Author Response · Authors · 2022-11-17
> > > > > **Reply**
> > > > >
> > > > > Dear reviewer, we have revised the manuscript in accordance with the detailed feedback that you have provided.

---

> > > > > > ### Comment · Reviewer_tGLV · 2022-11-29
> > > > > > **most of the concerns are resolved**
> > > > > >
> > > > > > I have gone through the author's response and most of the concerns are resolved.
> > > > > >
> > > > > > #4.3. The definition, although promised, is not added, at least around (8).
> > > > > >
> > > > > > #4.4. If the feature functions are selected as "any bounded feature function", how to guarantee that the derived function in (3) is exactly the covariance function of data?

---

> > > > > > > ### Author Response · Authors · 2022-11-29
> > > > > > > **Clarification on covariance model misspecification**
> > > > > > >
> > > > > > > Dear reviewer,
> > > > > > >
> > > > > > > Many thanks for your prompt feedback.
> > > > > > >
> > > > > > > > #4.3. The definition, although promised, is not added, at least around (8).
> > > > > > >
> > > > > > >  For clarity, we chose to gather all notational remarks, including the weighted norm in (8) in the introduction under *Notation:*. We apologize for not pinpointing the location of this revision in our reply above.
> > > > > > >
> > > > > > > > #4.4. If the feature functions are selected as "any bounded feature function", how to guarantee that the derived function in (3) is exactly the covariance function of data?
> > > > > > >
> > > > > > > The issue of *model misspecification* was addressed in point (ii) just below eq. (7): “performance guarantees that hold even when the model class [viz. (4)] is not well-specified”. That is, our results remain valid, *whether or not* the covariance model for (3) matches exactly the unknown covariance structure of the data.
> > > > > > >
> > > > > > > Based on the reviewer’s follow-up question, we have chosen to elaborate further on this point at the end of Section 2 (p.2). We hope that this amendment – along with the references to misspecified nominal covariance models in the abstract, Section 4.2 and 5.1 – will clarify the very *weak* modelling assumptions that we impose.

---

> > > > > > > > ### Comment · Reviewer_tGLV · 2022-11-29
> > > > > > > > **agree to accept**
> > > > > > > >
> > > > > > > > All of my concerns have been addressed. I would like to recommend acceptance once the updated manuscript is submitted.

---

### Decision · Action_Editors · 2022-12-20

**Recommendation:** Accept as is

**Comment:**

While one of the reviewer complains about the lack of novelty in the paper, the other two found the derived results are worth publication. I follow their suggestions here.
While the contributions are indeed limited, the paper is a nice extension and continuation of previous works by some of the authors that may be of interest to the machine learning community.

**Audience:**

This paper analyzes a covariance-fitting method inspired by (Stoica et al., 2010a;b) and (Zachariah & Stoica, 2015) to solve online prediction problems using linear smoothers. The authors show that this method may be implemented efficiently and comes with interesting theoretical features regarding out-of-sample error and robustness. Therefore, in my opinion, the paper is a  reasonable contribution to the field and should be of interest for TMLR's readers.



**Claims And Evidence:**

The contributions of the paper are well supported: the proofs of the main results appear to be clear and  the claims are illustrated with a limited but relevant numerical illustration.